# Robust Offloading for Edge Computing-Assisted Sensing and Communication Systems: A Deep Reinforcement Learning Approach

**DOI:** 10.3390/s24082489

**Published:** 2024-04-12

**Authors:** Li Shen, Bin Li, Xiaojie Zhu

**Affiliations:** 1School of Computer Science, Nanjing University of Information Science and Technology, Nanjing 210044, China; 202183290493@nuist.edu.cn; 2Division of Computer Science, King Abdullah University of Science and Technology, Thuwal 23955-6900, Saudi Arabia; xiaojie.zhu@kaust.edu.sa

**Keywords:** integrated communication and sensing, mobile edge computing, deep reinforcement learning, robust design, computation uncertainty

## Abstract

In this paper, we consider an integrated sensing, communication, and computation (ISCC) system to alleviate the spectrum congestion and computation burden problem. Specifically, while serving communication users, a base station (BS) actively engages in sensing targets and collaborates seamlessly with the edge server to concurrently process the acquired sensing data for efficient target recognition. A significant challenge in edge computing systems arises from the inherent uncertainty in computations, mainly stemming from the unpredictable complexity of tasks. With this consideration, we address the computation uncertainty by formulating a robust communication and computing resource allocation problem in ISCC systems. The primary goal of the system is to minimize total energy consumption while adhering to perception and delay constraints. This is achieved through the optimization of transmit beamforming, offloading ratio, and computing resource allocation, effectively managing the trade-offs between local execution and edge computing. To overcome this challenge, we employ a Markov decision process (MDP) in conjunction with the proximal policy optimization (PPO) algorithm, establishing an adaptive learning strategy. The proposed algorithm stands out for its rapid training speed, ensuring compliance with latency requirements for perception and computation in applications. Simulation results highlight its robustness and effectiveness within ISCC systems compared to baseline approaches.

## 1. Introduction

Recent years have witnessed a rise in intelligence applications and services. Integrated sensing and communication (ISAC) has been suggested as a pivotal concept in next-generation wireless communication systems [1]. Conventional methodologies separate sensing and communication, giving rise to challenges such as intricate design, bandwidth interference, and resource inefficiencies. Nevertheless, wireless sensing shares significant similarities with wireless communication technology in aspects such as hardware infrastructure and signal processing, making the mentioned integration possible. In view of the shared spectrum resources and hardware in ISAC systems, efficient resource utilization as well as mutual reciprocity and benefit between sensing and communication functions can be enabled [2].

However, with the implementation of these advanced functionalities, especially during the rapid growth of the internet of things (IoT), network edge nodes have begun to generate substantial amounts of data, thereby escalating the demand for effective data processing capabilities. Confronted with such voluminous data, devising a strategy for its rapid and efficient processing has emerged as an urgent and critical challenge [3]. In addressing this challenge, mobile edge computing (MEC) emerges as an innovative computing paradigm [4]. In this framework, proximally located servers are designated as edge servers, endowing user devices (UEs) with edge computing capabilities. This strategy markedly enhances data processing capabilities [5].

The network nodes in next-generation wireless communication systems will execute a variety of functions in an integrated manner, including high-precision, multi-objective environmental sensing and low-latency computing. This motivates the seamless integration of the ISAC network architecture with the MEC architecture, referred to as integrated sensing, communication, and computation (ISCC) [6], which is expected to support communication, sensing, and computation functionalities using the same signals and wireless infrastructure. Within this integrated framework, not only are the network nodes capable of simultaneously performing sensing and communication functions but also the system infrastructure is equipped to carry out efficient edge data processing [2]. To this end, ISCC can remarkably simplify equipment complexity and lower both production and usage costs significantly, which is essential for the advancement of wireless communication technology [7].

However, due to the unpredictability of computation types and the complexity of tasks, there is an inherent uncertainty in the computation processes of edge networks [8,9]. To address these challenges, a robust design scheme has been developed for the MEC network, which specifically accounts for the uncertainties associated with task complexity. To date, there has been limited research attention dedicated to the co-design of sensing, communication, and computation, especially when addressing the practical challenge of computation uncertainty. The main focus of this paper is to bolster a system’s robustness while simultaneously minimizing system energy consumption. The primary technical contributions of this paper can be outlined as follows:The computation uncertainty of a UE’s task within the framework of ISCC is investigated in this paper. To tackle the uncertainty of computation, a robust optimization problem aimed at minimizing system energy consumption is formulated. This is achieved by simultaneously optimizing communication and computation resources, beamforming, and offloading ratio.To address the outlined optimization challenges, this paper introduces a method that incorporates the proximal policy optimization (PPO) algorithm into deep reinforcement learning (DRL). This approach is designed to meet multiple constraints, including radar estimation information rate, computational offloading delay, and resource allocation. Utilizing a DRL training framework, this method allows for the efficient exploration and resolution of this intricate optimization problem. The system not only addresses practical constraints but also elevates decision making through the integration of intelligent learning algorithms.Through a series of simulation experiments, we assess the performance of this method, confirming the effectiveness of the computational robustness design and the PPO method in enhancing system efficiency and reducing energy expenditure. The robustness design demonstrates the improved performance in scenarios with uncertain task complexities. The simulations further reveal that the system’s weighted energy consumption could be significantly lowered when using the PPO algorithm with robustness.

The remainder of this paper is structured in the following manner. In Section 2, related work is detailed. In Section 3, the system model is proposed. Then, the training framework is advanced, and its complexity is analyzed in Section 4. Section 5 delivers detailed simulations to confirm the effectiveness and robustness of the algorithm. Finally, Section 6 draws conclusions.

## 2. Related Works

There have been many works focused on enhancing the performance of MEC systems. For instance, the authors of [10] suggested a resource allocation strategy using DRL, which adaptively allocates computation and network resources. This method aimed to decrease the mean service duration and balance resource usage under dynamic MEC conditions. Reference [11] investigated the minimization of system overhead in an MEC environment by jointly optimizing sampling, sensing, and computation offloading processes, effectively updating the status information of IoT systems. The work in [12] focused on optimizing resource allocation in ambient intelligence to maximize the convergence speed of federated edge learning. Further, ref. [13] presented the robust offloading policy and the joint allocation of communication and computation resources in an MEC system.

In the research on ISCC networks, efforts mainly place emphasis on the resource scheduling and optimization of beamforming techniques, ensuring the stability of communication links and enhancing the efficiency of computing task processing. In [14], the optimization of wireless spectrum resource strategies was performed in ISCC for dealing with high data transmission demands and complex computing tasks. The authors of [15] introduced adaptive digital twin technology in ISCC networks to improve network performance, especially in dynamic environments, in terms of application efficiency and reliability. While interesting, the works in [16,17] emphasized the significant role of beamforming in enhancing spectrum efficiency and reducing communication delay. The authors of [18] addressed the interference between radar sensing and MEC by jointly optimizing the sensing of beam pattern and task offloading with the aid of intelligent reflective surfaces. Considering previous work, there has been little research on robustness issues within ISCC networks. In this context, we study the computation uncertainty present in ISCC networks.

## 3. System Model and Problem Formulation

This section describes the system model for a robust ISCC system first and then formulates the energy consumption minimization problem.

### 3.1. System Model

This paper focuses on a beamforming and resource optimization issue in networks integrating communication, sensing, and computation, which are augmented by MEC. As shown in Figure 1, the network comprises one base station (BS) outfitted with *M* antennas, which serves *K* UEs with a single antenna each. The BS is not only equipped with an MEC server to enable computational offloading but also integrated with a radar sensing system, designed for the real-time detection and precise localization of potential targets. Concurrently, it ensures the provision of stable and reliable communication services to users. In practical applications, user devices can be regarded as smart wearable devices or AR devices, etc.

Taking into account the operational cycle and real-time requirements of the system, this paper posits that the task cycle is *T*, which is further subdivided into *N* time slots to facilitate meticulous resource scheduling, management, and computation. Consequently, the duration of each individual time slot is $\delta_{n}=T/N$, and it is assumed that the intervals between time slots are sufficiently short to satisfy the requirements for real-time processing. To simplify the problem, it is further assumed that within each time slot, the UE’s computing tasks, data transmission, and other operations can be completed within that same time slot.

Within this framework, the primary entities and time units of the network are defined. In this context, the sets of UEs are denoted as $\forall k\in K\overset{\Delta }{\;=}\{1,\ldots ,K\}$, and time slots are denoted as $\forall n\in N\overset{\Delta }{\;=}\{1,\ldots ,N\}$. Without the loss of generality, this paper adopts the Cartesian coordinate system. Each UE is associated with a specific two-dimensional coordinate $\left(x_{k},y_{k}\right)$, and the BS is assigned a fixed two-dimensional coordinate $\left(x_{\mathrm{BS}},y_{\mathrm{BS}}\right)$ and has a height of 100 meters. It should be noted that all UEs are located on the ground; therefore, only their two-dimensional coordinates are taken into consideration.

### 3.2. Signal Model

In time slot *n*, for BS and UE *k*, the communication channel is characterized as a Gaussian channel.

(1) Received Signal

During time slot *n*, the signal x[n], captured by the BS, is the superposition of the UE transmission signal $x_{\mathrm{com}}\left[n\right]$, the radar sensing signal $x_{\mathrm{sen}}\left[n\right]$, and the noise n[n], denoted as
(1)$$x\left[n\right]=x_{\mathrm{com}}\left[n\right]+x_{\mathrm{sen}}\left[n\right]+n\left[n\right]$$
where $n\left[n\right]\in C^{U\times 1}$ is an independent and identically distributed Gaussian random noise vector with a mean value of zero and a variance of $\sigma^{2}$.

The transmission signal from the UE and the radar sensing signal are the essential components of the signal received at the BS. The ensuing sections will delve into the detailed composition and properties of these transmission signals.

The transmission signal $x_{\mathrm{com}}\left[n\right]$ is characterized within the same time slot *n*, and the BS receives superimposed transmission signals from *K* distinct UEs, which can be expressed as
(2)$$x_{\mathrm{com}}\left[n\right]=\sum_{k\in K}x_{\mathrm{com}}^{k}\left[n\right]$$
where $x_{\mathrm{com}}^{k}\left[n\right]$ is used as the transmission signal of the UE *k*, expressed as
(3)$$x_{\mathrm{com}}^{k}\left[n\right]=p_{k}\left[n\right]s_{k}\left[n\right]h_{k}\left[n\right]$$
here, $p_{k}\left[n\right]$ is the transmission power allocated to UE *k* during time slot *n*, and $s_{k}\left[n\right]$ is the data symbol.

For the sensing signal $x_{\mathrm{sen}}\left[n\right]$, it is noted that the BS receives echoes from the target during time slot *n*. To effectively capture such echoes, the BS must first predict the radar’s emission signal based on prior knowledge of the target. Nonetheless, the signal that the radar emits may encounter various interferences on its return. To mitigate these interferences and more accurately extract information about the target, this paper adopts the approach of subtracting the radar’s transmission signal from the received signal to isolate the radar signal post-interference elimination, which can be described as ${\tilde{s}}_{\mathrm{sen}}\left[n\right]$. The beamforming vector $w_{\mathrm{sen}}\left[n\right]$ is then applied to ${\tilde{s}}_{\mathrm{sen}}\left[n\right]$ to process and obtain the resultant sensing received signal as follows:(4)$$x_{\mathrm{sen}}\left[n\right]=w_{\mathrm{sen}}\left[n\right]H_{\mathrm{sen}}\left[n\right]{\tilde{s}}_{\mathrm{sen}}\left[n\right]$$
where $w_{\mathrm{sen}}\left[n\right]\in C^{U\times 1}$ and $H_{\mathrm{sen}}\left[n\right]\in C^{U\times U}$ describe the target response matrix of the radar.

Furthermore, upon receiving the signal x[n], the BS employs the beamforming vector $w_{k}\left[n\right]$ to retrieve the signal. The retrieved signal is presented as
(5)$$\begin{array}{cc} \; & {\hat{x}}_{k}\left[n\right]=w_{k}^{H}\left[n\right]x\left[n\right]\\ \; & \;\;=w_{k}^{H}\left[n\right]\left(x_{\mathrm{com}}\left[n\right]+x_{\mathrm{sen}}\left[n\right]+n\left[n\right]\right)\\ \; & \;\;=w_{k}^{H}\left[n\right](\sum_{k=1}^{K}\sqrt{p_{k}\left[n\right]}h_{k}\left[n\right]s_{k}\left[n\right]\\ \; & \;\;+H_{\mathrm{sen}}\left[n\right]w_{\mathrm{sen}}\left[n\right]{\tilde{s}}_{\mathrm{sen}}\left[n\right]+n\left[n\right]) \end{array}$$

(2) Offloading Rate

Through beamforming, the directivity of the signal is optimized, enhancing the reception of the intended signal and concurrently attenuating the influence of unrelated signals and noise. This optimization is vital for the performance of communication systems, particularly for signal recovery and the data rates of UEs. Accordingly, in time slot *n*, the Shannon formula is employed to compute the offloading rate for the UEs, which can be articulated as follows:(6)$$R_{k}\left[n\right]=B\cdot \log_{2}\left(1+P_{s/n}\left[n\right]\right)$$
where *B* denotes the channel bandwidth, $P_{s/n}\left[n\right]$ represents the signal–noise power ratio in time slot *n*, which is the ratio of the signal power $s_{k}\left[n\right]$ to the noise power $n_{k}\left[n\right]$ during this interval. The signal power $s_{k}\left[n\right]$ and the noise power $n_{k}\left[n\right]$ are described as follows, respectively:(7)$$s_{k}\left[n\right]=p_{k}\left[n\right]{\left|w_{k}^{H}\left[n\right]h_{k}\left[n\right]\right|}^{2}$$
(8)$$\begin{array}{cc} n_{k}\left[n\right] & =B^{2}\psi^{2}\sigma_{\mathrm{sen}}^{2}{\left|w_{k}^{H}\left[n\right]H_{\mathrm{sen}}\left[n\right]w_{\mathrm{sen}}\left[n\right]\right|}^{2}\\ & +\sigma^{2}w_{k}^{H}\left[n\right]w_{k}\left[n\right]+\sum_{i=1,i\neq k}^{K}p_{i}\left[n\right]{\left|w_{k}^{H}\left[n\right]h_{i}\left[n\right]\right|}^{2} \end{array}$$
where, specifically, ψ represents the constant of the power amplifier, and $\sigma_{\mathrm{sen}}^{2}$ is the variance of the radar’s received signal noise.

### 3.3. Sensing Model

In the Signal Model section, a detailed description of the signal transmission process between the UEs and the BS is provided, as well as the specific expressions for each signal component. Nevertheless, it is also necessary to conduct the thorough modeling and analysis of the radar sensing aspect. The efficacy of radar sensing is intrinsically connected to the system’s capacity for the precise and efficient execution of tasks related to target detection and localization. Consequently, the key metric of radar estimation information rate is introduced as a solution for quantifying the signal’s sensing abilities [19].

The radar estimation information rate serves as the metric for target information acquired by radar. It is essentially the information shared between radar and the target through mutual interaction, quantifying the efficacy with which ISAC devices discern target information from the received echoes. This rate is employed to measure the volume of target information extractable from the sensing echoes.

In the context of radar estimation information rate, the signal–noise ratio of the radar echo signal is a key concept. It is denoted using a specific symbol $P_{r}\left[n\right]$ to denote the signal–noise ratio of the radar echo signal suppressed by ISAC devices in time slot *n*. The ratio of the radar duty cycle factor δ to the radar pulse duration μ, denoted by $B_{r}$, characterizes the proportion of time where the radar is actively transmitting versus the duration of a single pulse. These are described as follows:(9)$$P_{r}\left[n\right]=\frac{B^{2}\psi^{2}\sigma_{\mathrm{sen}}^{2}{\left|c{\left[n\right]}^{H}H_{\mathrm{sen}}\left[n\right]w_{\mathrm{sen}}\left[n\right]\right|}^{2}}{\sigma^{2}c{\left[n\right]}^{H}c\left[n\right]}$$
(10)$$B_{r}=\frac{\delta }{\mu }$$
where $c\left[n\right]\in C^{U\times 1}$ denotes the finite impulse response filter (FIR). Thus, during time slot *n*, the radar estimation information rate can be formulated as
(11)$$R_{r}\left[n\right]=\frac{1}{2}B_{r}\log_{2}\left(1+2B\mu P_{r}\left[n\right]\right)$$

### 3.4. Computation Model

In real-world application scenarios, task complexities often vary across different types. This paper has developed a multi-task model comprising a set of diverse task types, denoted as ∀z∈Z≜{1,…,Z}. The term $d_{k}\left[n\right]$ is used to quantify the data volume generated by UE *k* in time slot *n*, while $c_{z}$ indicates the intricacy linked to task *z*, reflecting the computational power required for processing. Given that the exact complexity $c_{z}$ may be unknown, computational uncertainty is introduced, reflecting the unpredictability of the real world. This can render the task scale measurable yet leave the completion time indeterminate. Through the analysis of historical data on multiple tasks, this research estimates the complexity of $c_{z}$, relating it to the error bound $\Delta \delta_{z}$. This error bound $\Delta \delta_{z}$ is constrained within a predefined threshold $\varepsilon_{z}$ to bolster robustness, which is represented as
(12)$$c_{z}={\hat{c}}_{z}+\Delta \delta_{z},\left|\Delta \delta_{z}\right|\leq \varepsilon_{z}$$

Furthermore, to facilitate the representation of task scheduling in time slot *n*, it is imperative to match the tasks and their respective types to each user independently, thereby ensuring that the task types assigned to each user within the same time slot are non-interfering. This is described as follows:(13)$$\begin{array}{cc} \; & \zeta_{z,k}\left[n\right]=\zeta_{z}\left[n\right]\zeta_{k}\left[n\right]\\ \; & \forall z\in Z,\zeta_{z}\left[n\right]\in \left\{0,1\right\}\\ \; & \forall k\in K,\zeta_{k}\left[n\right]\in \left\{0,1\right\} \end{array}$$

If $\zeta_{z}\left[n\right]=1$, then the assigned task type is *z*; conversely, if $\zeta_{z}\left[n\right]=0$, the task type is not *z*. In the same way, $\zeta_{k}\left[n\right]=1$ denotes that the task comes from UE *k*; otherwise, it does not. It is established that a task is attributed to UE *k* and classified as type *z* if and only if $\zeta_{z}\left[n\right]=1$ and $\zeta_{k}\left[n\right]=1$, a condition that can be succinctly described as $\zeta_{z,k}\left[n\right]=1$.

Due to the computational resource and energy constraints of UEs, it is infeasible to complete tasks locally within an expected timeframe. Consequently, this paper employs a partial offloading model. This means that each computational task is segmented into two components based on the offloading ratio $\rho_{k}\left[n\right]$. One proposed local computation is $d_{k}^{\mathrm{loc}}\left[n\right]$, and the other one is transferred to BS $d_{k}^{\mathrm{off}}\left[n\right]$, represented as follows:(14)$$d_{k}^{\mathrm{loc}}\left[n\right]=\left(1-\rho_{k}\left[n\right]\right)d_{k}\left[n\right]$$
(15)$$d_{k}^{\mathrm{off}}\left[n\right]=\rho_{k}\left[n\right]d_{k}\left[n\right]$$

(1) Delay

In the time slot *n*, the delay due to local computation for UE *k* is expressed as
(16)$$t_{k}^{\mathrm{loc}}\left[n\right]=\frac{\sum_{z=1}^{Z}d_{k}^{\mathrm{loc}}\left[n\right]\zeta_{z,k}\left[n\right]c_{z}}{f_{k}^{\mathrm{loc}}\left[n\right]}$$
where $f_{k}^{\mathrm{loc}}\left[n\right]$ denotes the processing rate of UE *k* in time slot *n*.

The offloading delay of a task offloaded from the user device to BS is described as
(17)$$t_{k}^{\mathrm{off}}\left[n\right]=\frac{d_{k}^{\mathrm{off}}\left[n\right]}{R_{k}\left[n\right]}$$

When the task from UE *k* is uploaded to the BS, since the BS is furnished with one MEC server, the MEC server processes the tasks submitted by the UE *k*, and the computational delay incurred by the MEC is denoted as
(18)$$t_{k}^{\mathrm{mec}}\left[n\right]=\frac{\sum_{z=1}^{Z}d_{k}^{\mathrm{off}}\left[n\right]\zeta_{z,k}\left[n\right]c_{z}}{f_{k}^{\mathrm{mec}}\left[n\right]}$$
where $f_{k}^{\mathrm{mec}}\left[n\right]$ denotes the processing rate by MEC for the task from UE *k*.

Given that the data volume processed by the MEC server is typically minimal, the delay associated with the return transmission is considered negligible in comparison to offloading and computational delays. Therefore, it is postulated that the return transmission occurs instantaneously. Consequently, the overall service delay for UE *k* is described as
(19)$$t_{k}^{fin}\left[n\right]=\max\left\{t_{k}^{\mathrm{off}}\left[n\right]+t_{k}^{\mathrm{mec}}\left[n\right],t_{k}^{\mathrm{loc}}\left[n\right]\right\}$$

(2) Energy Consumption

For the energy consumption attributable to computation, given that the energy resources of the BS can be considered unlimited, it is only necessary to account for energy consumption associated with the computation of UE *k*, which can be articulated as follows:(20)$$E_{k}^{\mathrm{loc}}\left[n\right]=\sum_{z=1}^{Z}\varepsilon {\left(f_{k}^{\mathrm{loc}}\left[n\right]\right)}^{2}d_{k}^{\mathrm{loc}}\left[n\right]\zeta_{z,k}\left[n\right]c_{z}$$
where ε is defined as the effective capacitance coefficients that depend on the chip architecture of the local computing device.

Beyond the energy consumed for computation, the energy expenditure for offloading transmissions also should be taken into consideration. As the return transmission is assumed to be instantaneous and the data volume is approximated to zero, the energy cost associated with the return transmission is deemed negligible. Therefore, only the energy consumed during offloading is considered. The offloading energy from UE *k* is articulated as follows:(21)$$E_{k}^{\mathrm{off}}\left[n\right]=\frac{d_{k}^{\mathrm{off}}\left[n\right]}{R_{k}\left[n\right]}p_{k}\left[n\right]$$

Therefore, in time slot *n*, UE *k*’s energy consumption can be obtained:(22)$$E_{k}\left[n\right]=E_{k}^{\mathrm{off}}\left[n\right]+E_{k}^{\mathrm{loc}}\left[n\right]$$

### 3.5. Problem Formulation

This study endeavors to minimize the aggregate system energy consumption throughout the entire cycle by jointly optimizing the offloading ratio $\rho ≜\{\rho_{k}\left[n\right],\forall k\in K,n\in N\}$, the computational resource allocation of the MEC to each UE $f_{e}≜\left\{f_{k}^{\mathrm{mec}}\left[n\right],\forall k\in K,n\in N\right\}$, the local computing resource allocation for UEs $f_{l}≜\left\{f_{k}\left[n\right],\forall k\in K,n\in N\right\}$, and the beamforming $W≜\{w_{k}\left[n\right],\forall k\in K,n\in N\}$. The optimization problem is denoted as follows: (23)$$\begin{array}{cc} \; & \;\;\;\;\;\;\;\;C0:\;\min_{\left\{\rho ,f_{e},f_{l},W\right\}}\;\sum_{n\in N}\sum_{k\in K}E_{k}\left[n\right]\\ \; & \;\;s.t.\;C1:\;0\leq \rho_{k}\left[n\right]\leq 1,\forall k\in K,n\in N\\ \; & \;\;\;\;\;C2:\;\zeta_{z,k}\left[n\right]\in \left\{0,1\right\},\forall z\in Z,k\in K,n\in N\\ \; & \;\;\;\;\;C3:\;\sum_{z\in Z}\zeta_{z,k}\left[n\right]=1,\forall k\in K\\ \; & \;\;\;\;\;C4:\;0\leq f_{k}^{\mathrm{mec}}\left[n\right]\leq f_{\mathrm{mec}}^{\max},\forall k\in K,n\in N\\ \; & \;\;\;\;\;\;\;\;\;\;\;\sum_{k=1}^{K}f_{k}^{\mathrm{mec}}\left[n\right]\leq f_{\mathrm{mec}}^{\max},\forall k\in K,n\in N\\ \; & \;\;\;\;\;C5:\;0\leq f_{k}^{\mathrm{loc}}\left[n\right]\leq f_{k}^{\max},\forall k\in K,n\in N\\ \; & \;\;\;\;\;C6:\;0\leq p_{k}\left[n\right]\leq P_{k}^{\max},\forall k\in K,n\in N\\ \; & \;\;\;\;\;C7:\;t_{k}^{fin}\left[n\right]\leq t^{\max},\forall k\in K,n\in N\\ \; & \;\;\;\;\;C8:\;\left|\delta_{z}\right|\leq \varepsilon_{z},\forall z\in Z\\ \; & \;\;\;\;\;C9:\;R_{r}\left[n\right]\geq R_{r}^{\min},\forall n\in N\; \end{array}$$
where $f_{\mathrm{mec}}^{\max}$ and $f_{k}^{\max}$ signify the maximum processing rates of the MEC server and UE *k* for tasks, respectively, while $P_{k}^{\max}$ denotes the maximum transmitting power of UE *k*, and $t^{\max}$ constrains the slot time for a single task. The term $\Delta \delta_{z}$ is bounded within a radius of $\varepsilon_{z}$, and $R_{r}^{\min}$ represents the minimum radar estimation information rate. Constraint C1 pertains to the proportion of the task offloaded to the MEC server. C2 and C3 describe the type of task generated by UE *k*. C4 limits the maximum computational resources assigned to UE *k* by the MEC server. C5 restricts UE *k*’s computational resources. C6 governs the offloading transmission power of UE *k*. C7 delineates the task duration. C8 manages the computational error. C9 prescribes the lower bound for the minimum performance of radar perception.

## 4. Proposed Algorithm

### 4.1. Modeling of Single-Agent MDP

A Markov decision process (MDP) is employed to model the challenge in this paper, which involves a nonlinear objective function and environmental uncertainties. To address the challenges posed by time-varying channels and multitasking, DRL is utilized to discover optimal strategies. Despite the increased complexity due to high-dimensional state spaces and the synchronization delay, the PPO variant of DRL algorithms is adopted. PPO ensures stable and effective policy learning for a single agent in complex environments and is instrumental in optimizing resource allocation and task execution within MEC networks to achieve minimal energy consumption. The training framework is shown in Figure 2.

MDP provides a potent mathematical framework to tackle optimization problems [20]. In the MDP paradigm, a decision maker strategically selects actions from a defined set of states. Each action precipitates a state transition coupled with an associated immediate reward. The decision maker’s goal is to orchestrate a sequence of actions that amplifies the expected cumulative reward from the present state to a designated future juncture. MDP is especially applicable to contexts where environmental predictability is limited. Through the strategic application and resolution of MDP, the most advantageous policy sequence can be ascertained, offering a theoretically optimal solution to intricate decision-making problems.

State space: To holistically account for the attributes of tasks and the resources of the BS, the state space is defined as
(24)$$\begin{array}{cc} s_{n}= & \{L_{1}\left[n\right],L_{2}\left[n\right],\ldots ,L_{k}\left[n\right]\;;\\ \; & C_{1}\left[n\right],C_{2}\left[n\right],\ldots ,C_{k}\left[n\right]\} \end{array}$$
where the set of tasks is described by $L\left[n\right]=\left[L_{1}\left[n\right],L_{2}\left[n\right],\ldots ,L_{k}\left[n\right]\right]$, and UEs’ CPU resources are $C\left[n\right]=\left[C_{1}\left[n\right],C_{2}\left[n\right],\ldots ,C_{k}\left[n\right]\right]$.

Action space: The agent, upon receiving the state space $s_{n}$, determines actions represented by $a_{n}$, which dictate the task offloading choices and the allocation of resources at the BS, thereby quantifying the resultant utility. Therefore, the action can be expressed as
(25)$$a_{n}=\left\{\rho \left[n\right],f_{e}\left[n\right],w\left[n\right]\right\}$$

Simultaneously, with the goal of minimizing the energy expended in user computations, this paper employs dynamic voltage frequency scaling technology to configure and estimate the CPU frequency, as detailed in the subsequent formula:(26)$$f_{k}\left[n\right]=\min\left\{\frac{d_{k}\left[n\right]\zeta_{z,k}\left[n\right]c_{z}}{t_{k}^{fin}\left[n\right]},f_{k}^{\max}\right\}$$

Reward space: To encapsulate the long-term optimization objectives of the problem and to address the fulfillment of constraints, this paper devises a reward function analogous to system energy consumption. The reward encompasses the energy expenditure of the user as well as penalties incurred for breaches of delay and perception constraints. Consequently, the reward function formulated in this paper is illustrated as follows:(27)$$r_{n}=-\left[\sum_{k=1}^{K}E_{k}\left[n\right]\right]P_{n}^{\mathrm{sen}}P_{n}^{T}$$

Among them, the perception constraint penalty $P_{n}^{\mathrm{sen}}$ is a linear penalty function, and the delay constraint penalty $P_{n}^{T}$ is an exponential penalty function, described in Equations (28) and (29), respectively:(28)$$P_{n}^{\mathrm{sen}}\;=\;1\;\;+\;\left(\frac{{\bar{R}}_{\mathrm{sen}}-R_{\mathrm{sen}}^{\min}}{R_{\mathrm{sen}}^{\max}-R_{\mathrm{sen}}^{\min}}\right)$$
(29)$$\begin{array}{cc} \; & P_{n}^{T}=\frac{1}{K}\sum_{k\in K}P\left(t_{k}^{fin}\left[n\right],T_{k}\left[n\right]\right)\\ \; & =\frac{1}{K}\sum_{k\in K}\left(2-\exp(-{\left\lceil \left(t_{k}^{fin}\left[n\right]-T_{k}\left[n\right]\right)/T_{k}\left[n\right]\right\rceil }^{+})\right) \end{array}$$
in which ${\left\lceil \cdot \right\rceil }^{+}$ means that the value is rounded up.

### 4.2. PPO-Based DRL Training Framework

To address the problem presented in this paper, PPO, a sophisticated policy gradient technique in DRL, is utilized within the MDP framework to optimize problem-solving methods. PPO is distinguished for its efficiency and capacity for robust optimization in policy spaces, garnering widespread popularity. This algorithm is characterized by a balanced approach to exploration and exploitation, achieved through limiting policy update steps. Such a mechanism is crucial in reducing variance during training and enhancing learning stability. This method is especially beneficial for optimization problems that demand continuous decision making and encompass a broad parameter space, such as dynamic system control and complex resource management tasks. Unlike algorithms that adopt an off-policy approach, the PPO algorithm is an on-policy method, meaning that the behavior policy and target policy are the same. This allows the algorithm to converge more quickly. Based on the on-policy method, agents trained with the PPO algorithm can continuously improve their policies while interacting with the environment, making it easier to adapt to changes in the environment while maintaining exploratory behavior. The effective learning of complex strategies, without compromising performance, is facilitated by PPO, thus providing more precise and robust solutions to the optimization problems.

Within the proposed framework, the actor–critic architecture plays a crucial role. The actor network is responsible for making decisions, essentially determining the action to be taken given the current state. Conversely, the critic network evaluates these actions by estimating the value function, providing feedback on the quality of the decisions made by the actor. In this architecture, the actor network is divided into new and old components, corresponding to parameters θ and $\theta_{\mathrm{old}}$, respectively. Furthermore, the critic network corresponds to network parameter ς. This collaborative adjustment ensures that the policy update is directed towards the enhancement of expected returns.

In the actor–critic framework, the state value function $V(s_{n})$ becomes a pivotal component for strategy refinement. It encapsulates the expected returns from the current state $s_{n}$ under the policy π, providing a crucial metric for the strategic adjustment process. The specific mathematical expression is as follows:(30)$$\begin{array}{cc} \; & V\left(s_{n}\right)=E\left\{R(a_{n}|s_{n}),\pi \right\}\\ \; & \;\;\;\;\;\;\;\;=E\left\{\sum_{l=0}^{\infty }\gamma^{l}R\left(a_{n+l}|s_{n+l}\right)\right\} \end{array}$$
where E denotes the expected value operator, and γ represents the discount factor for future rewards, indicating the relative importance of future rewards compared to immediate ones. R symbolizes the reward obtained from a given state and action pair.

The action value function, represented as $Q(s_{n},a_{n})$, is a critical tool for evaluating the expected return of selecting an action $a_{n}$ in a given state $s_{n}$ and adhering to a policy π for subsequent actions. It comprehensively accounts for the immediate rewards and the aggregated impact of potential future rewards. The action value function may be indicated as
(31)$$Q\left(s_{n},a_{n}\right)=E\left\{\sum_{l=0}^{\infty }\gamma^{l}R\left(a_{n+l}|s_{n+l}\right)\right\}$$

Building on this foundation, the advantage function is introduced to assess the efficacy of the selected actions within the given policy framework.
(32)$$A\left(s_{n}\right)=Q\left(s_{n},a_{n}\right)-V\left(s_{n}\right)$$

To guarantee the stability of policy updates, the framework employs the general advantage estimation (GAE) approach. The estimated advantage function $\hat{A}\left(s_{n}\right)$ is denoted as
(33)$$\hat{A}\left(s_{n}\right)=\sum_{l=0}^{\infty }{\left(\gamma \lambda \right)}^{l}\left(r_{n+l}+\gamma V\left(s_{n+l+1}\right)-V\left(s_{n}\right)\right)$$
where λ is the GAE coefficient. Subsequently, the evaluation network and the action network are updated by optimizing the ensuing objective function. The critic network ς and actor network θ can be renewed by employing the following function:(34)$$G\left(\varsigma \right)={\left[V\left(s_{n+1}\right)-V\left(s_{n}\right)\right]}^{2}$$
(35)$$\begin{array}{cc} \; & G\left(\theta \right)=E\left\{\min\left[\frac{\pi_{\theta }\left(a_{n}|s_{n}\right)}{\pi_{\theta_{\mathrm{old}}}\left(a_{n}|s_{n}\right)}\hat{A}\left(s_{n}\right),\right.\right.\\ \; & \;\;\;\;\left.\left.\mathrm{clip}\left(\frac{\pi_{\theta }\left(a_{n}|s_{n}\right)}{\pi_{\theta_{\mathrm{old}}}\left(a_{n}|s_{n}\right)},1-\varepsilon ,1+\varepsilon \right)\hat{A}\left(s_{n}\right)\right]\right\} \end{array}$$
where $\pi_{\theta }$ and $\pi_{\theta_{\mathrm{old}}}$ represent the new and old policy functions, respectively, and the update ratio is denoted by $\frac{\pi_{\theta }\left(a_{n}|s_{n}\right)}{\pi_{\theta_{\mathrm{old}}}\left(a_{n}|s_{n}\right)}$. The introduction of the clipping factor ε serves to constrain the policy’s update ratio, ensuring controlled and stable optimization steps.

Algorithm 1 presents the pseudocode for the DRL training procedure utilizing the PPO algorithm.
**Algorithm 1** Proposed PPO training framework1:Initialize the maximum training episodes $l_{m}$, the maximum episode length $l_{e}$, the PPO epochs $l_{p}$, the BS’s location $\left(x_{BS},y_{BS}\right)$.2:Initialize Critic network ς, Actor network θ.3:**for** $m\in \{1,\ldots ,l_{m}\}$ **do**4:   Initialize users’ location $\left(x_{k},y_{k}\right)$ and users’ tasks5:   **for** $n\in \{1,\ldots ,l_{e}\}$ **do**6:     Obtain state $s_{t}$ from the environment7:     Make decisions $\pi_{\theta }$ from the state $s_{t}$8:     Choose action $a_{n}$ based on $\pi_{\theta }$9:     Execute action $a_{n}$ and update to next state $s_{n+1}$10:     Calculate rewards $r_{n}$11:     Store experience $\left(s_{n},a_{n},r_{n},s_{n+1}\right)$12:   **end for**13:   **for** $n\in \{1,\ldots ,l_{e}\}$ **do**14:     calculate $\hat{A}\left(s_{n}\right)$15:   **end for**16:   **for** $n\in \{1,\ldots ,l_{p}\}$ **do**17:     update ς and θ18:   **end for**19:**end for**

### 4.3. Complexity Analysis

The complexity of the PPO algorithm, as proposed by the Computing Institute, is calculated in this section. Algorithm 1’s computational complexity is gauged by the count of multiplication operations executed in a single iteration. Within the DRL framework, the observed state values are initially transmitted to a multi-layer perceptron (MLP) by the agent. A typical MLP consists of an input layer, an output layer, and multiple hidden layers. The state values enter the MLP through the input layer, are processed through the hidden layers, and they are ultimately outputted by the output layer. Given that the input layers as well as output layers have little impact on performance, they are typically disregarded. Consequently, the complexity of each layer can be characterized as follows:(36)$$O\left(N_{i-1}N_{i}+N_{i}N_{i+1}\right)$$
where $N_{i}$ represents the quantity of neurons in a specific hidden layer. Thus, the computation complexity of the layer I MLP is as follows:(37)$$O\left(\sum_{i=2}^{I-1}\left(N_{i-1}N_{i}+N_{i}N_{i+1}\right)\right)$$

Both the actor and critic networks are constituted by an MLP. Based on the previous analysis, the total computational complexity of Algorithm 1 is, thereby, deduced as follows:(38)$$O\left(l_{m}\left(l_{e}\left(\sum_{i=2}^{I-1}\left(N_{i-1}N_{i}+N_{i}N_{i+1}\right)\right)\right)\right)$$

Further, we reference the scheme proposed in [21], which addresses a scenario similar to ours, employing a WMMSE algorithm for optimizing resource allocation along with sensing and communication beamforming. The study in [22] introduces a beamforming framework enhanced by an LSTM network to augment communication efficiency, this method has also applied to our scenario for comparative analysis. The complexity comparisons of these three algorithms are systematically presented in Table 1.

For this comparison, we assume a setup with 16 users and 4 antennas. Considering that the learning algorithm incorporates convolutional layers, we set the maximum number of iterations to 300 and have 2 intermediate convolutional layers. The analysis clearly demonstrates that our algorithm is better. The complexity of the WMMMSE algorithm and the complexity of the LSTM algorithm are 10 and 2 times higher, respectively, compared to the proposed algorithm.

## 5. Simulation Results

This section defines the simulation data to verify the impact of the MEC-assisted ISCC network based on the PPO algorithm on the system’s overall energy use. The simulation environment is constructed using the PyTorch framework. A thorough assessment of the performance of the proposed solution is conducted as follows.

### 5.1. Parameter Settings

The number of BSs is set to one, with the BS fixed at the coordinates (0,0) and positioned at an elevation of 100 m. Consider the initiation of activities within a terrestrial square region with a specified area of 1000m×1000m. The users are randomly distributed within this area. Task data sizes are uniformly distributed in $\left[L_{\min},\;L_{\max}\right]$, with $L_{\min}$ defaulting to 0.5Mb and $L_{\max}$ defaulting to 1.5Mb. The mean number of cycles per bit is $C_{k}\left[n\right]\in \left[500,1500\right]$ cycles/bit. The duration of the task cycle, denoted as *T*, is set as 200s. Unless explicitly stated otherwise, the default configuration comprises 16 users, with the communication channel bandwidth between the users and the BS being preset to B=10MHz. The noise-related power levels, denoted by parameters $\eta^{2}$ and $\eta_{sen}^{2}$, are set as −70dBm uniformly. Some of the default parameters for the environmental settings are delineated in Table 2, while parameters pertinent to algorithmic training are enumerated in Table 3.

### 5.2. Simulation Evaluation

To validate the algorithm’s performance, this study will conduct the following comparative assessments:

(1) Baseline PPO Design: This approach utilizes the PPO algorithm, devoid of enhancements for computation robustness.

(2) Computational Robust Design: This variant integrates robust computational design, leveraging the DRL-enhanced PPO algorithm introduced herein, which seeks to determine nearly optimal actions for the ensuing time increment.

(3) Complete Offloading: This scenario entails the wholesale relegation of tasks to the BS for execution, thereby absolving users from computational duties.

(4) Random Offloading: In this model, users offload tasks to the BS in a stochastic manner, retaining the responsibility to compute any portion of the tasks not designated to the BS.

The convergence of the PPO algorithm is first substantiated through the analysis presented in Figure 3a. It is clear that with a growing number of training steps, the reward associated with the proposed solution correspondingly ascends. The enhancement of the stated reward by the agent is significant, which substantiates the great performance of the PPO algorithm in the context of computational offloading. Employing PyTorch for the computational experiments, an ensemble of data was amassed through 60,000 training steps, where each result is the sum of reward values within a round. Throughout the training phase, the strategies employed by the agent, particularly in communication, perception, and computation, undergo optimization. Concomitantly, the observed performance fluctuations become smaller, eventually allowing the algorithm to stabilize at the stable reward threshold.

In order to examine the impact of learning rate on algorithmic convergence, a comparative analysis of reward value convergence curves at different learning rates was undertaken. As depicted in Figure 3a, the learning rate of $5\times 10^{-3}$ achieves convergence at approximately 2000 training steps, while a learning rate of $5\times 10^{-4}$ requires about 8000 training steps for convergence. Additionally, when the learning rates are set at $2\times 10^{-3}$ and $8\times 10^{-4}$, the convergence of the curves is observed to lie between the rates mentioned above. Despite the variability in convergence steps necessitated by different learning rates, once convergence was achieved, the resultant reward values were found to be insignificantly varied, residing within a stable range. These findings elucidate that the learning rate exerts a discernible effect on the PPO algorithm’s convergence velocity, yet its influence on performance efficacy remains marginal.

Figure 3b provides a clear comparative analysis of the performance between the PPO method with the robust design computations and the non-robust method. The computationally robust PPO algorithm, as proposed in this paper, is observed to achieve higher and more stable overall rewards compared to the baseline PPO algorithm. This improved performance is largely attributable to the computationally robust strategies.

Figure 4 provides an in-depth comparative analysis of the changes in system-weighted energy consumption for four distinct methods across varying user scales. It is observed that the PPO algorithm, designed with a focus on computational robustness as proposed in this paper, outperforms others in terms of efficiency. The system’s average weighted energy consumption under various user sizes is found to be lower when employing this algorithm, in contrast to the baseline PPO approach. Moreover, it is noted that the energy consumption resulting from strategies such as complete offloading or random offloading is significantly higher, which substantiates the effectiveness of incorporating partial offloading strategies based on offloading ratio ρ in the joint optimization process, thereby effectively reducing system energy consumption and enhancing performance. Additionally, an upward trend in average energy consumption between adjacent user numbers is observed in Figure 4. This increase is attributed to the growing number of users accessing the network, which escalates signal interference among users, reduces transmission rates, and consequently raises transmission costs. Such developments lead to a decrease in the volume of tasks offloaded to BS and an increase in locally computed tasks, necessitating greater computational resources from users and ultimately resulting in a rise in system energy consumption.

Figure 5a and Figure 5b show the average weighted energy consumption of users under varying computational task sizes and different bandwidth settings, respectively. In Figure 6, with the minimum task size set at $L_{\min}=0.5\;\mathrm{Mb}$, it is observed that user energy consumption incrementally rises with an increase in the maximum task size $L_{\max}$, whereas energy consumption diminishes with the expansion of bandwidth. Similarly, in Figure 5b, with the maximum task size established at $L_{\max}=4.0\;\mathrm{Mb}$, it is noted that user energy consumption escalates as the minimum task size $L_{\min}$ increases. Furthermore, energy consumption intensifies as the available bandwidth narrows. This phenomenon can be attributed to the fact that the augmentation of communication resources enhances the users’ transmission rates, whereas the escalation in computational tasks results in an upsurge in users’ average computation energy consumption. This is because more sizable tasks necessitate a greater allocation of computational resources, thereby leading to increased energy expenditure. Moreover, the data depicted in the graphics indicate that with the increment in bandwidth, a divergence in energy consumption emerges under various bandwidth conditions. This primarily originates from the fact that, for the BS, an expanded bandwidth implies a reduction in transmission latency, and the shortened transmission time is afforded to BS for the processing of more computational tasks. The increase in bandwidth incentivizes users to offload a greater number of tasks to the BS, thus alleviating the local computational workload. Hence, the enhancement of bandwidth conserves computational CPU resources for users, thereby effectively reducing the computation energy consumption.

Figure 6 shows the influence of estimation error bounds and task complexity on performance. It is evident from the figure that when the estimation error bound increases, the energy consumption rises. This is attributed to the fact that a bigger error bound results in better uncertainty in the calculations.

## 6. Conclusions

In this article, the computation uncertainties of tasks in ISCC systems was investigated, leading to the robust offloading and resource allocation scheme. By jointly optimizing transmit beamforming, offloading factors, communication and computation resource allocation, the system energy consumption minimization problem was formulated. To effectively address this optimization challenge, a PPO framework was developed to facilitate the efficient implementation of optimal learning policies. Extensive numerical results have highlighted the superiority of the proposed scheme in terms of energy consumption reduction as compared with baseline approaches. In future research, further investigations into resource allocation and optimization decisions in ISCC networks will be pursued by considering various task processing environments.

## Figures and Tables

**Figure 1 sensors-24-02489-f001:**
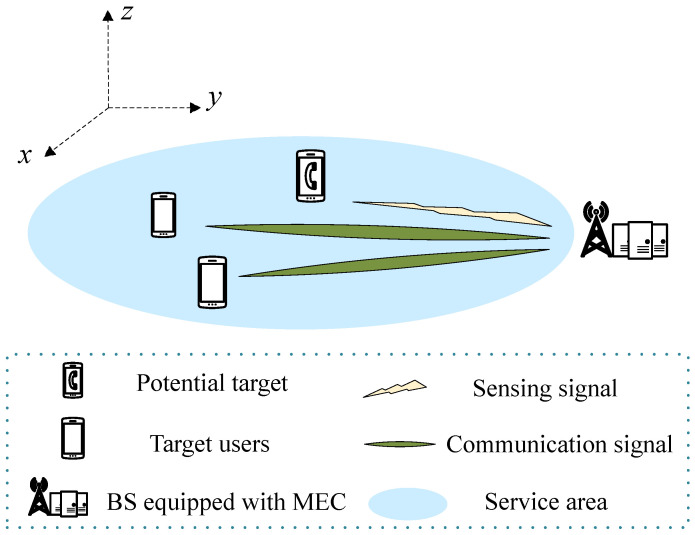
System model.

**Figure 2 sensors-24-02489-f002:**
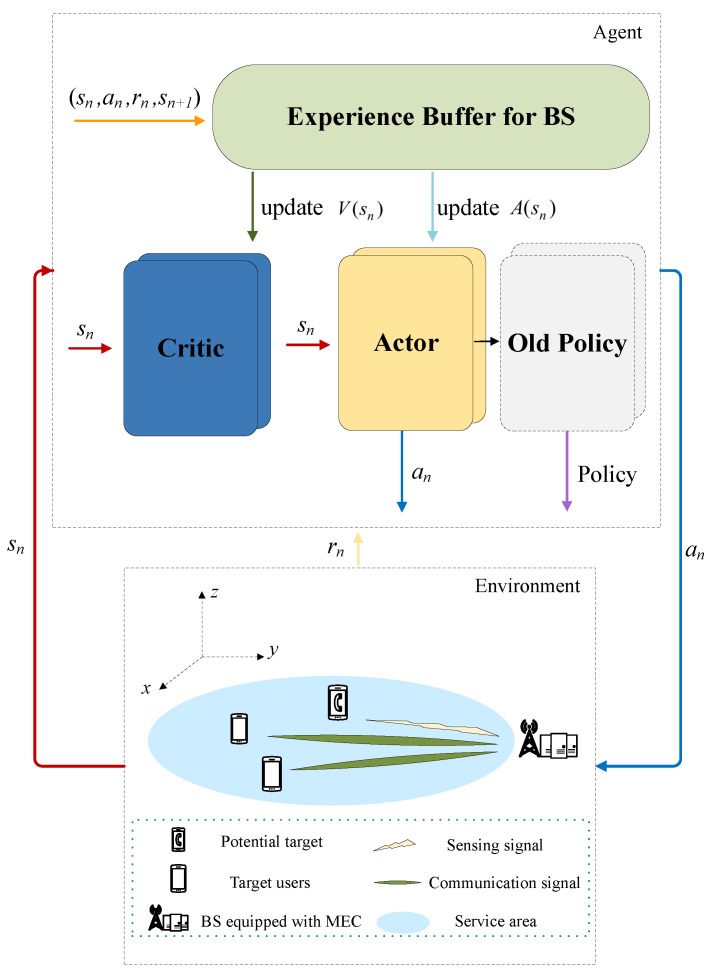
Training framework.

**Figure 3 sensors-24-02489-f003:**
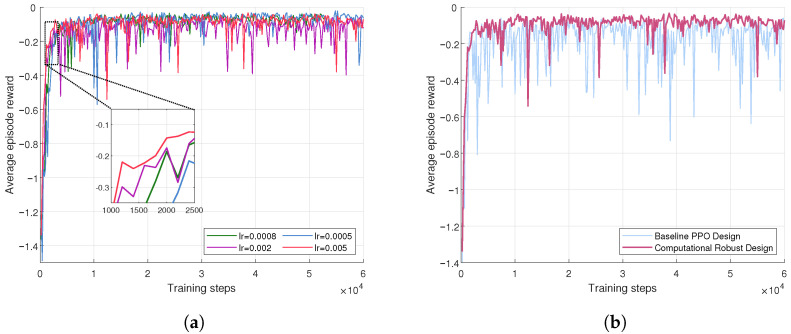
Performance differences in different trainings steps. (**a**) Convergence with different learning rate; (**b**) comparison with robust and non-robust.

**Figure 4 sensors-24-02489-f004:**
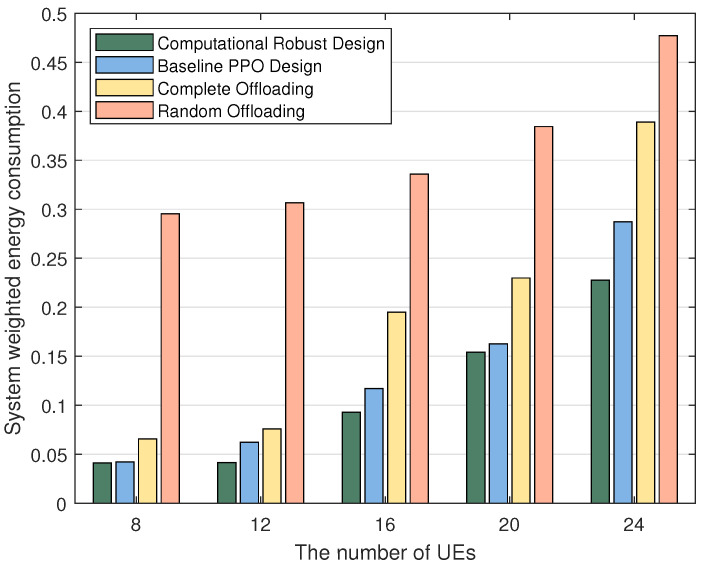
Performance comparison between different numbers of UEs.

**Figure 5 sensors-24-02489-f005:**
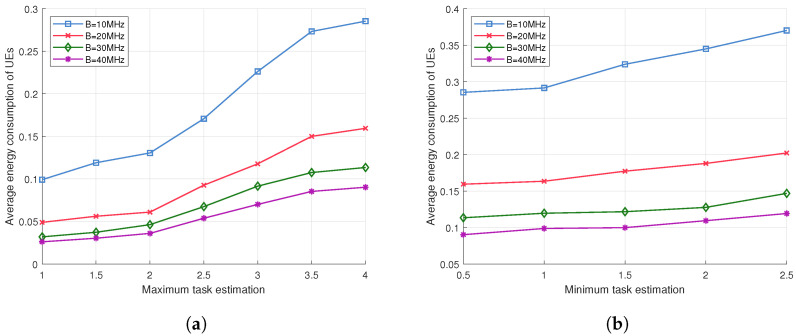
Energy consumption in different bandwidths and task estimation. (**a**) Performance comparison of different maximum task volumes and bandwidth; (**b**) performance comparison of different minimum task volumes and bandwidth.

**Figure 6 sensors-24-02489-f006:**
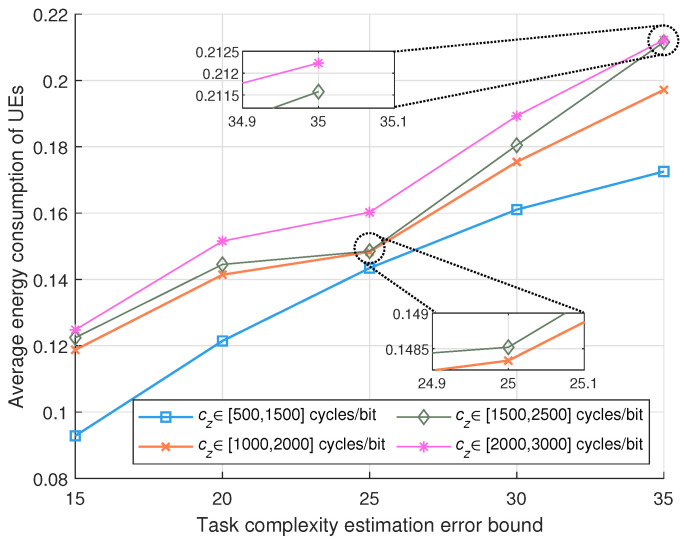
Performance comparison of different task volumes and task complexity estimation error bounds.

**Table 1 sensors-24-02489-t001:** Comparison of computational complexity of different algorithms.

Algorithm	Computational Complexity
WMMMSE algorithm [21]	$O(l_{m}(2K^{2}W^{3}+2K^{3}$ $+K^{1/2}\left(4K+W\right){\left(3K+W\right)}^{2}+6K^{2}))$
LSTM algorithm [22]	$O(4l_{m}\cdot N_{e}\cdot l_{e}\left(G_{1}G_{2}+G_{2}^{2}+G_{2}\right))$
Proposed algorithm	$O\left(l_{m}\left(l_{e}\left(\sum_{i=2}^{I-1}\left(N_{i-1}N_{i}+N_{i}N_{i+1}\right)\right)\right)\right)$

**Table 2 sensors-24-02489-t002:** Environment settings.

Parameters	Values
Time slot $\delta_{n}$	1.0 s
Constant ψ	$\pi /\sqrt{3}$
Radar duty cycle factor δ	0.01
Radar pulse duration μ	$2\times 10^{-5}$ s
Predefined threshold $\varepsilon_{z}$	15
MEC maximum frequency $f_{\mathrm{mec}}^{\max}$	8 GHz
UE maximum frequency $f_{k}^{\max}$	1.5 GHz
UE maximum transmitting power $p_{k}^{\max}$	0.4 W
Minimum radar estimation information rate $R_{\mathrm{rad}}^{\min}$	$10^{3}$ dB
Effective capacitance coefficients ε	$10^{-27}$

**Table 3 sensors-24-02489-t003:** Training settings.

Parameters	Values
Learning rate lr	$5\times 10^{-3}$
Maximum training set $l_{m}$	300episodes
Length of each training set $l_{e}$	200steps
Discount factor γ	0.95
GAE parameters λ	0.96

## Data Availability

The data used to support the findings of this study are included within the article.

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
