# Peer review of "Robust Offloading for Edge Computing-Assisted Sensing and Communication Systems: A Deep Reinforcement Learning Approach"

_sensors, 2024, doi:10.3390/s24082489_

Round 1

Reviewer 1 Report

Comments and Suggestions for Authors

This paper considers an Integrated Sensing, Communication and Computation (ISCC) system to alleviate the spectrum congestion and computation burden problem. The authors provide a wealth of data to support the proposed views. However, the following major concerns should be addressed.

1. The motivations of this paper are not clearly expressed. Specifically, they identify the challenges of problem-solving. However, they don't explain the differences from existing work and the shortcomings of existing work. Related works are only listing the existing work without critically analyzing the pitfalls in existing work. Several paragraphs in sections 1 and 2 needs to be added. It should focus on explicitly highlighting where the gap in the literature is located and how can the proposed solution address that gap. Therefore, sections 1 and 2 should be revised.

2. For Fig. 1, one or two illustrative examples/practical applications should be added. In addition, could you provide more explanation or insight on Fig. 1?

3. The benchmark schemes that the authors compare are with conventional schemes, which may be unfair for algorithm performance evaluation. I strongly suggest that the authors should compare the proposed algorithm with the current works.

4. The setting of simulation parameters seems a little unreasonable. For example, why is the UE maximum transmitting power set to 0.4W? In general, 0.2W is a common value. Is 8GHz and 1.5GHz reasonable for computing resources? The author needs to provide more references. In addition, why are the training parameters these values?

5. The author needs to discuss the optimality or performance loss of the algorithm.

Author Response

Please view the attached response file.

Reviewer 2 Report

Comments and Suggestions for Authors

Article is dedicated to solving the problem of optimal management of communication resources. The theme of the article is relevant. The structure of the article slightly differs from that preferred by MDPI for research articles (Introduction (including analog analysis), Models and Methods, Results, Discussion, Conclusions). The level of English is acceptable. The article is easy to read. The figures in the article are of acceptable quality. The article cites 19 sources, some of which are not current. The article made a positive impression on me. However, regarding the content of the article, I would like to make the following comments and recommendations:

1. I kindly ask the authors to specify the parametric features of "Edge Computing-Assisted Sensing." How is the specificity of this process reflected in the parametric space of the optimization problem?

2. The optimization problem (23) does not seem complicated. Why apply a "Deep Reinforcement Learning Approach" to solve a relatively trivial optimization problem? Operations research methods can solve such optimization problems and will be more computationally efficient than any "Deep" approach in any case.

3. In the experimental section, I did not see justification for the adequacy of the proposed approach through comparison of benchmark results with simulation results.

4. I believe that in future research, authors should consider that many of the parameters they mention are dynamic rather than static characteristics. Moreover, ignoring this fact provokes inaccuracies in the proposed approach, the magnitude of which should be estimated.

5. Finally, authors should link their model to current technologies of real communication platforms, such as 5G. Currently, everything looks too abstract, which raises doubts about the practical value of the obtained result.

Author Response

Please view the attached response file.

Reviewer 3 Report

Comments and Suggestions for Authors

This article investigates an edge computing-assisted ISAC system to address the weighted energy consumption where robust offloading is designed to improve performance. The authors propose a proximal policy optimization algorithm and extensive simulations are provided to validate the superior performance of the proposed method. It sounds an interesting and timely work. However, I have the following comments/suggestions for further improvements.

1. Why is deep reinforcement learning used for the solution and what problem does the deep reinforcement learning method overcome?

2. Does the occurrence of excessive individual energy consumption among users emerge when considering system optimization? In case so, would this impact the optimization process? If not, provide why such occurrences are unlikely.

3. In Section 4(Proposed Algorithm), the state value function and the action value function are mentioned. So how to balance the importance of current and future rewards? How to ensure that both long-term and short-term returns are considered in the decision-making process?

4. Several grammatical and spelling errors in the article, e.g. formula (23), and the context should be consistent. I suggest the authors proofread the article to further improve the quality.

5. Some related references should be cited as follows:

Do***S, **a Y, Kamruzzaman J. Quantum particle swarm optimization for task offloading in mobile edge computing[J]. IEEE Transactions on Industrial Informatics, 2022.

Comments on the Quality of English Language

ok

Author Response

Please view the attached response file.

Round 2

Reviewer 1 Report

Comments and Suggestions for Authors

All my questions have been addressed carefully, and I have no more questions.

Reviewer 3 Report

Comments and Suggestions for Authors

The paper has addressed all my questions.